# Survival benefits of perioperative chemoradiotherapy versus chemotherapy for advanced stage gastric cancer based on directed acyclic graphs

**Cheng Zheng[1]☯, Yue Zhang[1]☯, Juan Cao[1], Xiaoying Jing[1], HongHui Li[1,2]***

1 Department of Epidemiology and Statistics, School of public health and management, Ningxia Medical University, Yinchuan, Ningxia, China, 2 Department of Occupational and Environmental Health, School of Public Health and Management, Ningxia Medical University, Yinchuan, China

☯ These authors contributed equally to this work.
* h297268657@163.com

**Data Availability Statement:** All relevant data are contained within the paper. Additional information can be obtained by contacting Dr. Honghui Li (h297268657@163.com).

## Abstract

The overall survival benefits of perioperative chemotherapy (PCT) and perioperative chemoradiotherapy (PCRT) for patients with locally advanced gastric cancer (GC) have not been fully explored. The aim of this study was to compare the benefits of PCT and PCRT in GC patients and determine the factors affecting survival rate using directed acyclic graphs (DAGs). The data of 1,442 patients with stage II-IV GC who received PCT or PCRT from 2000 to 2018 were retrieved from the Surveillance, Epidemiology, and End Results (SEER) database. First, the least absolute shrinkage and selection operator (LASSO) was used to identify possible influencing factors for overall survival. Second, the variables that were selected by LASSO were then used in univariate and Cox regression analyses. Third, corrective analyses for confounding factors were selected based on DAGs that show the possible association between advanced GC patients and outcomes and evaluate the prognosis. Patients who received PCRT had longer overall survival than those who received PCT treatment ($P = 0.015$). The median length of overall survival of the PCRT group was 36.5 (15.0 − 53.0) months longer than that of the PCT group (34.6 (16.0 − 48.0) months). PCRT is more likely to benefit patients who are aged $\leq$ 65, male, white, and have regional tumors ($P<0.05$). The multivariate Cox regression model showed that male sex, widowed status, signet ring cell carcinoma, and lung metastases were independent risk factors for a poor prognosis. According to DAG, age, race, and Lauren type may be confounding factors that affect the prognosis of advanced GC. Compared to PCT, PCRT has more survival benefits for patients with locally advanced GC, and ongoing investigations are needed to better determine the optimal treatment. Furthermore, DAGs are a useful tool for contending with confounding and selection biases to ensure the proper implementation of high-quality research.

**Funding:** This study was supported by the Natural Science Foundation of Ningxia (No. 2022AAC03173). The funders had role in study design, data collection and analysis, decision to publish, or preparation of the manuscript.

**Competing interests:** The authors have declared that no competing interests exist.

## Introduction

Gastric cancer (GC) remains a prevalent human malignant disease and the fourth leading cause of cancer-related death globally [1]. In the United States, approximately 11,140 patients died from this disease in 2019 [2]. In Western countries, more than half of patients have a locally advanced stage at initial diagnosis [3]. Because advanced-stage GC has largely heterogeneous characteristics, an uncertain mechanism, limited therapy strategies, and a less than 12-month median survival time [3–5], it is essential to develop optimal individualized management strategies for such patients.

Gastrectomy remains the primary treatment for locally advanced GC [6]. However, satisfactory outcomes cannot be achieved by surgical resection alone, and the five-year survival rate is only 20%- 50%, leading to efforts to improve the survival of these patients receiving neoadjuvant or adjuvant therapies [7]. The MAGIC trial and the FLOT4 trial were milestone studies that showed superior survival in patients who received perioperative chemotherapy (PCT) compared to surgery alone for GC [8–10]. In America, the use of PCT in patients with T2+ gastric adenocarcinoma in America increased from 34% in 2006 to 65% now, thus, becoming a standardized treatment for GC [11]. Taking into account the high local recurrence and metastasis rates in patients with GC, chemotherapy (CT) combined with radiotherapy (RT) has been proposed and compared with CT in several clinical trials. The INT0116 trial revealed significant survival benefits of adjuvant chemoradiotherapy (CRT) for GC patients after surgery [12]. Additionally, the results of the POET trial demonstrated that the inclusion of RT in preoperative treatment led to survival advantages [13]. On the other hand, the CRITICS trial showed that in patients who underwent preoperative CT, postoperative CRT did not improve survival when compared to postoperative CT [14]. The ARTIST trial also showed that adjuvant RT combined with CT did not have a positive influence on patient survival [15]. A growing body of evidence suggests that the careful assessment of the survival benefits of adjuvant and neoadjuvant CRT is necessary for treatment strategy selection. Until now, evidence comparing PCT and PCRT has been limited to several small, randomized trials that have been conducted in East Asian countries rather than North American countries [16–18]. In Europe and the United States, it remains unclear whether PCT should be used in combination with radiotherapy and whether chemoradiotherapy is efficacious. Therefore, screening advanced patients to determine if they can receive PCRT is extremely critical to improving survival rates.

Determining causal relations and eliminating bias are the primary aims of research conducted by the scientific community. To date, observational research has primarily focused on relationships between covariates and outcomes, with authors only rarely asserting a direct causal relationship. Because randomization has been regarded as the most reliable strategy for eliminating bias among treatment groups, statements of direct causal relationships have only been made in the context of randomized trials. Although randomized clinical trials are useful for addressing bias, they are expensive, time-consuming, frequently unfeasible, and unrepresentative of the intended audience when blinding methods are used [19, 20]. Given that observational research has enormous unrealized potential to guide clinical decision-making, structured approaches to examine causal connections in specified datasets are being increasingly noticed [21]. Directed acyclic graphs (DAGs) are visual representations of causality in scientific discussions and are being increasingly used in modern epidemiology. Pearl and Spirtes popularized techniques of causal inference using DAGs in 1995 [22], but its use in dental research was only first advocated in 2002, and several subsequent studies have been directed toward theoretical exploration rather than clinical practice [23, 24]. This visual aid helps to explicitly describe the underlying relationships defined in the scientific discussion.

Therefore, the purpose of the current study was to compare the predicted length of survival of advanced GC patients who received PCT with that of patients who received PCRT. Furthermore, we screened features related to PCT and PCRT in patients with advanced GC and drew a DAG to identify and control potential confounders so that clinicians can select a more appropriate treatment strategy for these patients.

## Material and methods

### Population

The available data from a retrospective cohort study were extracted using SEER*stat (8.3.6) software. In our study, we used the SEER database's tumor nomenclature and coding manual [25] as well as the International Classification of Diseases tumor morphology code ICD-O-3 to extract data from GC patients treated between 2004 and 2018 [26]. All the data used in this study were openly accessible and retrieved from the SEER database. TNM staging was used to classify all cancer samples according to the American Joint Committee on Cancer (AJCC). Inclusion criteria: (1) patients who were diagnosed from 2004 to 2018 with stage II – IV; (2) patients in whom the primary site of GC was the stomach; (3) patients for whom the exact treatment strategy was PCT or PCRT; and (4) patients with a pathologically confirmed diagnosis of GC. Ineligible cases with unknown or missing characteristic data were excluded. S1 Fig shows the flowchart of patient selection. The study used deidentified data and adhered to the World Medical Association's Declaration of Helsinki for Ethical Human Research. Informed consent was obtained from all subjects and their legal guardian.

### Study variables

The following variables were extracted from the SEER cohort: sex, age, race, marital status, primary site, histologic type, T stage, N stage, M stage, tumor size, differentiation, summary stage, Lauren type, bone metastases, brain metastases, liver metastases, lung metastases, and comprehensive treatment. The information for metastatic sites of the bone or brain or liver and lung (SEER Combined Mets at DX- bone or brain or liver and lung) and comprehensive treatment (CT, RT, systemic therapy) were collected in 2010; thus, metastatic GC and systemically treated patients diagnosed with stage II–IV diseases from 2010 to 2015 were included. The continuous variable was transformed into a categorical variable, such as age ($\leq$ 65 and > 65 years old) and tumor size ($\leq$ 5 cm and > 5 cm). The summary stage was categorized into regional, distant, and localized. Primary sites were divided into five subsites as follows: cardiac and fundus, body, antrum and pylorus, lesser and greater curvature, and others. The tumors were pathologically categorized into poorly differentiated, moderately differentiated, well undifferentiated, and undifferentiated. The histological types were categorized into intestinal types, diffuse types, mixed types, and other types. The pathology types were divided into adenocarcinoma and signet ring cell. The primary outcome of the study was overall survival (OS), which was calculated from the date of diagnosis until the date of any cause of death or a follow-up termination event.

### Defining covariates for a directed acyclic graph

DAGs comprises a series of nodes representing variables and arrows representing causal relationships between different variables. The nodes are selected based on the prognostic factors included in the Cox proportional hazards model for functional outcomes in advanced GC patients. The arrow's direction is based on recent literature or a priori knowledge. To orient a series of arrows to facilitate causal interpretation, we applied the following prior knowledge-

based constraints: adjuvant concurrent CT or CRT is the standard care for patients with resected advanced GC [14]. Therefore, the primary survival outcomes were specified as a sink with only inward-pointing arrows, and the secondary outcome measure treatment including PCT and PCRT was specified as a source with only outward-pointing arrows. Moreover, many nonmodifiable variables, such as age, sex, race, marital status, histologic type, Lauren type, and lung metastases, are also related to advanced GC. A study pointed out a direct association between age and adenocarcinoma and intestinal-type metastasis. The clinicopathological characteristics showed that there were more metastatic diseases in the young patients and more intestinal types in the old patients, and the majority of them were male [27, 28]. Another study reported that age and sex may be modifiers of the effects of adjuvant CRT [29]. Furthermore, GC is a phenotypically highly heterogeneous disease that may exhibit a variety of biological behaviors, as patients with intestinal-type GC had better overall survival than those with diffuse-type and mixed-type GC [5, 28]. Patients exhibit different sensitivities to CT or CRT according to Lauren's classification [11, 30, 31]. Thus, it is difficult to select the optimum treatment. Marital status may also affect the prognosis in GC patients; in particular, marriage plays a positive role [32]. Studies also indicated that unmarried or widowed patients with various tumor types were at a high risk of metastatic presentation and had shorter survival [32–34]. By applying the tetrad software with these constraints to the information, the DAG model was generated.

## Statistical analyses

First, the LASSO was used to identify possible influencing factors for overall survival. The "glmnet" package was used to perform the Lasso regression model analysis. The baseline characteristics of the training set and the validation set, randomly divided by lasso regression, were analyzed using the $\chi^2$ test or Fisher's exact probability method. Second, the variables of statistical significance were selected by the LASSO from the training set and then used in the univariate analysis. Survival curves were estimated using the Kaplan–Meier method, and the log-rank test was used to determine survival differences between the groups. Third, independent risk factors that affected OS in advanced GC patients were determined in a Cox regression model. Relative risks were estimated by calculating the hazard ratio (HR) and 95% confidence intervals (CIs). Variables by multivariate analysis were incorporated into nomograms that were constructed as visual graphics. Finally, corrective analyses for confounding factors were selected based on DAGs that showed the possible association between advanced GC patients, primitive tumors, comprehensive treatment, and outcomes. The p values were derived from two-tailed tests, and p values< 0.05 was considered statistically significant. Statistical analysis was performed with SPSS software version 23.0 (SPSS, Inc., Microsoft, Chicago IL, USA) and R studio software (version 3.6.1; https://www.rstudio.com). The DAG was drawn by using tetrad 6.9.0 web-based software (Tools – Center for Causal Discovery (pitt.edu)).

## Results

### Patient's baseline characteristics (overall sample)

A total of 1422 advanced GC patients, including 289 females and 1153 males, were included in the current study, including 410 patients who received PCT and 1032 patients who received PCRT. After random sampling at a ratio of 2:1, 1022 and 420 patients were included in the training set and validation set, respectively. The mean age was 61±11 years. In most of the patients, 1174 patients (81.4%), the tumor was located at the cardiac or fundus, and regional tumors were reported in 1106 patients (76.7%). There were 822 patients with poor differentiation, accounting for 57.0% of all patients, and 1233 patients with an advanced T stage (T3-T4),

accounting for 85.5% of all patients. The characteristics of all the GC patients who met the inclusion and exclusion criteria are shown in S1 Fig, and the data of the patients in the two sets are presented in S1 Table.

## Feature selection

In all 18 associated characteristic variables, 16 potential predictors were considered from the cohort data (S2 Fig) and were retained with nonzero regression coefficients in the LASSO algorithm. K cross-validation for centralization and normalization of included factors was performed 10 times and then the best lambda value was chosen. The best tuning parameter lambda for the LASSO regression was 0.0018 when the partial-2 log-likelihood binomial deviance reached its minimum value. The area under the receiver operating characteristic (ROC) curve was used to provide good discrimination for the quality of the model by the lasso regression to separate true positives from false positives. Then, all the selected variables had significant differences and were applied to develop the nomogram models. The nomogram in this study only presents the independent risk factors in the multivariate analysis.

## Survival by treatment groups

To identify the prognostic factors related to overall survival, subgroup analyses stratified by comprehensive treatment were performed. The results demonstrated that the survival of the PCRT group was significantly better than that of the PCT group in the training set (Table 1). Patients who received PCT had a longer OS than those who received PCRT treatment (hazard ratio = 0.846, 95% CI = 0.738–0.970, P = 0.015). The median length of overall survival of the PCRT group was 36.5 (15.0 − 53.0) months longer than that of the PCT group (34.6 (16.0 − 48.0) months). In comparison to PCT, PCRT benefits patients who are aged ≤ 65, male, white, and have regional tumors (P<0.05). These results indicated that PCRT benefitted patients with advanced GC in terms of survival.

## Univariate and multivariate analyses

In the univariate analysis, the factors significantly associated with advanced GC were marital status, primary site, histology type, marital status, TNM stage, differentiation, summary stage, lung metastases, liver metastases, and comprehensive treatment (Fig 1). In the multivariate Cox regression model, male sex, widowed status, signet ring cell carcinoma, and lung metastases were considered to be independent risk factors for a poor prognosis (Fig 2). PCRT was still significantly associated with better survival in patients with advanced GC (HR = 0.862, 95% CI = 0.744–0.996, P = 0.044).

## Directed acyclic graph analysis

Corrective analyses for variables in the multivariate model were selected based on DAGs that showed the possible relationship with advanced GC patients (Fig 3). Based on our univariate and multivariate analyses, the following variables were included in the DAG analysis: sex, age, race, marital status, histologic type, Lauren type, lung metastases, and comprehensive treatment.

Although DAGs presented in previous literature can intuitively reflect the impact of each variable on the survival outcome of advanced GC patients, it is not clear which relationship is sufficiently meaningful, so it is essential to test the DAGs (Table 2). Five variables directly influenced the primary survival outcome: sex (r = -0.0676, HR = 1.032), marital status (r = 0.0268, HR = 1.021), histology type (r = 0.0565, HR = 1.015), lung metastases (r = 0.3200,

**Table 1. Univariate analysis of factors associated with the use of perioperative chemotherapy and perioperative chemoradiotherapy (n = 1442).**

| Characteristic | Total | Training set (n = 1022) | | | Validation set (n = 420) | | |
|---|---|---|---|---|---|---|---|
| | | PCT (n = 287) Mortality (%) | PCRT (n = 735) Mortality (%) | Pa | PCT (n = 123) Mortality (%) | PCRT (n = 297) Mortality (%) | P |
| Age (years) | | | | 0.010 | | | 0.500 |
| ≤ 65 | 576 (65.6) | 144 (75.0) | 265 (61.9) | 0.012 | 56 (70.0) | 111 (62.4) | 0.174 |
| > 65 | 389 (69.0) | 70 (73.7) | 202 (65.8) | 0.336 | 30 (69.8) | 87 (73.1) | 0.559 |
| Sex | | | | 0.006 | | | 0.549 |
| Male | 782 (67.8) | 159 (77.6) | 400 (64.6) | 0.017 | 57 (66.3) | 166 (68.3) | 0.484 |
| Female | 183 (63.3) | 55 (67.1) | 67 (57.8) | 0.192 | 29 (78.4) | 32 (59.3) | 0.011 |
| Race | | | | 0.007 | | | 0.347 |
| White | 807 (67.4) | 151 (75.9) | 424 (64.4) | 0.022 | 58 (69.9) | 174 (67.7) | 0.685 |
| Black | 65 (69.9) | 32 (78.0) | 14 (58.3) | 0.265 | 11 (78.6) | 8 (57.1) | 0.131 |
| Other | 93 (61.2) | 31 (66.0) | 29 (54.7) | 0.386 | 17 (65.4) | 16 (61.5) | 0.701 |
| Marital status | | | | 0.019 | | | 0.564 |
| Married | 633 (65.0) | 143 (76.1) | 295 (59.7) | <0.001 | 58 (69.0) | 137 (65.9) | 0.602 |
| Divorced/Separated | 114 (72.6) | 24 (80.0) | 59 (73.7) | 0.981 | 10 (71.4) | 21 (63.6) | 0.727 |
| Single | 136 (67.3) | 30 (66.7) | 68 (68.0) | 0.585 | 13 (72.2) | 25 (64.1) | 0.331 |
| Widowed | 53 (82.8) | 9 (64.3) | 30 (83.3) | 0.119 | 5 (100.0) | 9 (100.0) | 0.501 |
| Unknown | 29 (64.4) | 8 (80.0) | 15 (60.0) | 0.480 | 0 (0) | 6 (75.0) | 0.189 |
| Primary site | | | | 0.219 | | | 0.963 |
| Cardiac/fundus | 767 (65.3) | 115 (75.2) | 429 (62.8) | 0.078 | 42 (67.7) | 181 (65.6) | 0.635 |
| Body | 24 (70.6) | 10 (66.7) | 6 (85.7) | 0.240 | 2 (40.0) | 6 (85.7) | 0.022 |
| Antrum/pylorus | 54 (68.4) | 32 (71.1) | 8 (88.9) | 0.611 | 14 (60.9) | 0 (0) | 0.166 |
| Lesser/greater curvature | 53 (70.7) | 24 (72.7) | 7 (43.7) | 0.121 | 16 (84.2) | 6 (85.7) | 0.482 |
| Other | 67 (83.7) | 33 (80.5) | 17 (85.0) | 0.181 | 12 (85.7) | 5 (100.0) | 0.007 |
| Histology | | | | 0.051 | | | 0.654 |
| Adenocarcinoma | 753 (65.1) | 145 (75.1) | 385 (62.2) | 0.022 | 60 (65.2) | 163 (64.4) | 0.996 |
| Signet ring cell carcinoma | 155 (75.2) | 50 (75.8) | 62 (71.3) | 0.610 | 20 (83.3) | 23 (79.3) | 0.209 |
| Other | 57 (72.2) | 19 (67.9) | 20 (69.0) | 0.417 | 6 (85.7) | 12 (80.0) | 0.781 |
| TNM Stage | | | | 0.209 | | | 0.998 |
| II | 235 (53.3) | 35 (54.7) | 135 (52.1) | 0.582 | 10 (47.6) | 55 (56.7) | 0.362 |
| III | 644 (71.0) | 153 (78.5) | 295 (67.5) | 0.151 | 65 (73.0) | 131 (70.4) | 0.433 |
| IV | 86 (91.5) | 26 (92.9) | 37 (94.9) | 0.566 | 11 (84.6) | 12 (85.7) | 0.365 |
| T stage | | | | 0.189 | | | 0.985 |
| T1-2 | 126 (60.3) | 19 (61.3) | 72 (62.1) | 0.782 | 10 (62.5) | 25 (54.3) | 0.584 |
| T3 | 697 (66.2) | 142 (74.3) | 357 (62.4) | 0.027 | 43 (67.2) | 155 (68.4) | 0.66 |

*(Continued)*

**Table 1.** (Continued)

| Characteristic | Total | Training set (n = 1022) | | | Validation set (n = 420) | | |
|---|---|---|---|---|---|---|---|
| | | PCT (n = 287) Mortality (%) | PCRT (n = 735) Mortality (%) | Pa | PCT (n = 123) Mortality (%) | PCRT (n = 297) Mortality (%) | P |
| T4 | 142 (78.9) | 53 (81.5) | 38 (80.9) | 0.244 | 33 (76.7) | 18 (72.0) | 0.697 |
| N stage | | | | 0.962 | | | 0.787 |
| N0 | 178 (55.8) | 31 (64.6) | 100 (52.9) | 0.086 | 6 (60.0) | 41 (56.9) | 0.842 |
| N1 | 399 (62.0) | 56 (61.5) | 222 (60.7) | 0.602 | 34 (65.4) | 87 (64.4) | 0.926 |
| N2 | 261 (79.3) | 73 (84.9) | 116 (80.6) | 0.649 | 23 (69.7) | 49 (74.2) | 0.997 |
| N3 | 127 (84.7) | 54 (87.1) | 29 (80.6) | 0.728 | 23 (82.1) | 21 (87.5) | 0.372 |
| M stage | | | | 0.047 | | | 0.834 |
| M0 | 879 (65.2) | 188 (72.6) | 430 (61.8) | 0.021 | 75 (68.2) | 186 (65.7) | 0.611 |
| M1 | 86 (91.5) | 26 (92.9) | 37 (94.9) | 0.566 | 11 (84.6) | 12 (85.7) | 0.365 |
| Differentiation | | | | 0.050 | | | 0.865 |
| Poorly | 587 (71.4) | 145 (74.6) | 265 (68.2) | 0.318 | 58 (72.5) | 119 (73.5) | 0.718 |
| Moderately | 249 (59.3) | 47 (72.3) | 138 (57.7) | 0.257 | 15 (65.2) | 49 (54.4) | 0.150 |
| Well | 26 (57.8) | 3 (75.0) | 8 (40.0) | 0.214 | 2 (50.0) | 13 (76.5) | 0.174 |
| Undifferentiated | 103 (65.2) | 19 (82.6) | 56 (61.5) | 0.106 | 11 (68.7) | 17 (60.7) | 0.516 |
| Summary stage | | | | 0.020 | | | 0.599 |
| Regional | 724 (65.5) | 167 (73.9) | 341 (62.0) | 0.030 | 67 (65.7) | 149 (65.4) | 0.684 |
| Distant | 177 (79.7) | 40 (87.0) | 88 (76.5) | 0.331 | 18 (90.0) | 31 (75.6) | 0.896 |
| Localized | 64 (56.1) | 7 (46.7) | 38 (54.3) | 0.979 | 1 (100.0) | 18 (64.3) | 0.091 |
| Lauren type | | | | 0.019 | | | 0.637 |
| Intestinal | 47 (62.7) | 12 (80.0) | 20 (54.1) | 0.201 | 7 (70.0) | 8 (61.5) | 0.906 |
| Diffuse | 38 (73.1) | 17 (68.0) | 9 (69.2) | 0.287 | 6 (85.7) | 6 (85.7) | 0.901 |
| Mixed | 24 (75.0) | 9 (90.0) | 10 (71.4) | 0.726 | 3 (60.0) | 2 (66.7) | 0.854 |
| Unknown | 856 (66.7) | 176 (74.3) | 428 (63.8) | 0.019 | 70 (69.3) | 182 (66.4) | 0.633 |
| Tumor size (cm) | | | | 0.025 | | | 0.542 |
| ≤ 5 | 505 (65.0) | 92 (69.2) | 245 (61.9) | 0.411 | 43 (71.7) | 125 (66.5) | 0.232 |
| >5 | 270 (70.1) | 87 (82.1) | 114 (62.7) | 0.005 | 31 (67.4) | 38 (70.4) | 0.484 |
| Unknown | 190 (67.9) | 35 (72.9) | 108 (67.5) | 0.900 | 12 (70.6) | 35 (63.6) | 0.726 |
| Bone metastases | | | | 0.014 | | | 0.487 |
| Yes | 40 (65.6) | 6 (50.0) | 22 (61.0) | 0.093 | 6 (66.7) | 6 (66.7) | 0.873 |
| No/Unknown | 925 (67.0) | 208 (75.6) | 445 (63.2) | 0.004 | 80 (70.2) | 192 (66.7) | 0.499 |
| liver metastases | | | | 0.025 | | | 0.566 |
| Yes | 24 (92.3) | 7 (87.5) | 12 (100.0) | 0.115 | 2 (66.7) | 3 (100.0) | 0.405 |

*(Continued)*

**Table 1.** (Continued)

| Characteristic | Total | Training set (n = 1022) | | | Validation set (n = 420) | | |
|---|---|---|---|---|---|---|---|
| | | PCT (n = 287) Mortality (%) | PCRT (n = 735) Mortality (%) | Pa | PCT (n = 123) Mortality (%) | PCRT (n = 297) Mortality (%) | P |
| No/Unknown | 941 (66.5) | 207 (74.2) | 455 (62.9) | 0.010 | 84 (70.0) | 195 (66.3) | 0.486 |
| lung metastases | | | | 0.017 | | | - |
| Yes | 11 (100.0) | 4 (100.0) | 5 (100.0) | 0.717 | 2 (100.0) | 2 (100.0) | - |
| No/Unknown | 954 (66.7) | 210 (74.2) | 462 (63.3) | 0.015 | 86 (69.9) | 196 (66.4) | 0.490 |

Abbreviations: PCT, Perioperative chemotherapy; PCRT, Perioperative chemoradiotherapy. P for intra-group univariate survival analysis; Pa for inter-group univariate survival analysis.

HR = 1.024), and treatment (r = -0.0992, HR = 1.001). Four of these variables directly influenced the secondary outcome measure treatment: sex (r = -0.0676, HR = 1.001), race (r = -0.1314, HR = 1.001), histology type (r = -0.1032, HR = 1.001) and Lauren type (r = -0.0318, HR = 1.040). Sex, race, and Lauren type were associated with the primary survival outcome, but these factors were not associated with the secondary outcome measure treatment (Table 2, all P<0.05), which indicated that these may be confounding factors by comparing the results with the Cox regression analysis.

## Discussion

In this cohort study, the SEER database was used to retrospectively analyze the demographic and clinical characteristics of patients with locally advanced GC to evaluate the effect of PCT

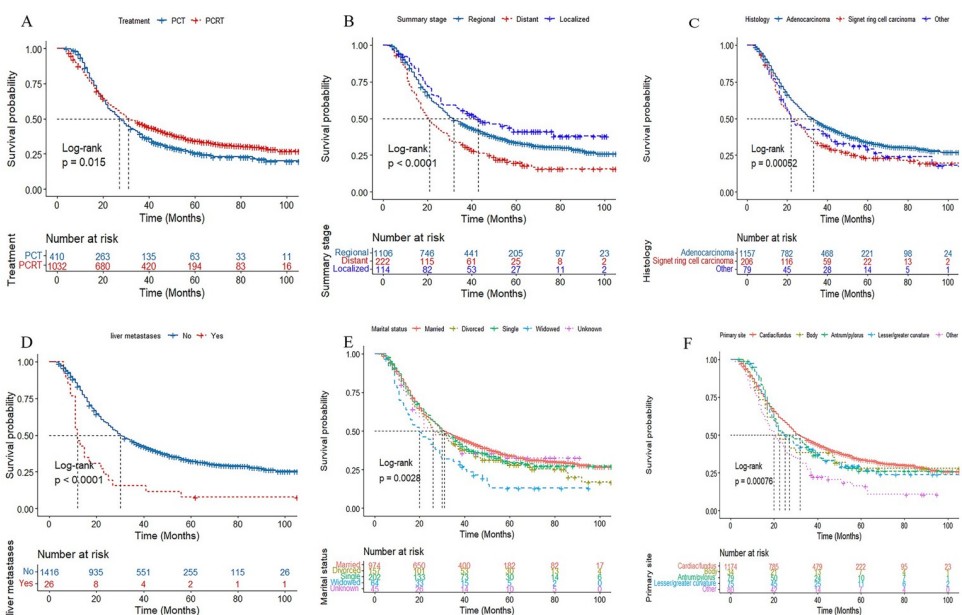

**Fig 1. Kaplan-Meier survival curves of clinically related factors in advanced stage gastric patients.** (A-F) The dotted lines indicate the median survival time of patients. Overall survival for patients with (A) treatment, (B) summary stage, (C) histology, (D) live metastases, (E) marital status, and (F) primary site.

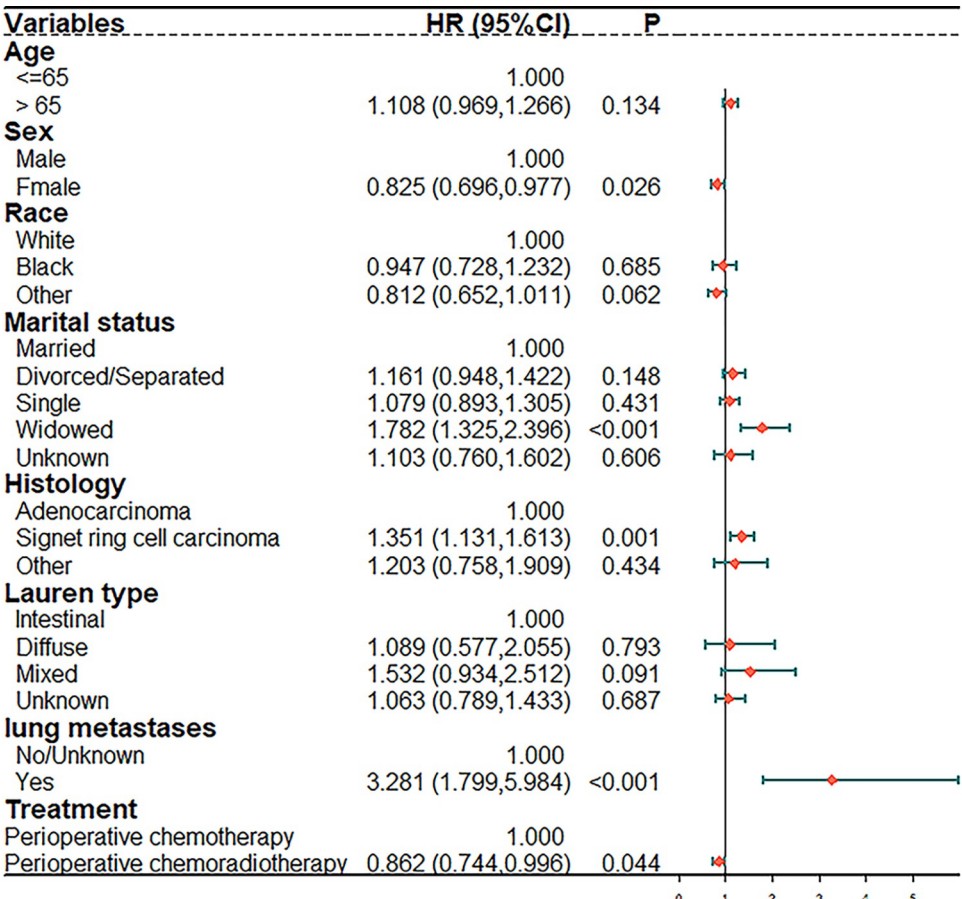

**Fig 2. Multivariable cox regression analysis of the clinical features in the overall population.** Forest plot demonstrating gastric cancer with advanced stage hazard ratios.

and PCRT on the prognosis of such patients. The results indicated that PCRT had significant survival benefits when compared to PCT for patients with advanced GC. In a study using the SEER database, 21,447 patients with stage I-IV GC benefited most from adjuvant RT with CT when compared with surgery or PCT [35]. In the multivariate Cox regression model, male sex, widowed status, signet ring cell carcinoma, and lung metastases were considered to be independent risk factors for a poor prognosis. Sex, race, and Lauren type were associated with the primary survival outcome, but these factors were not associated with the secondary outcome measure treatment by DAG methods, which indicated that these may be confounding factors by comparing the results with the Cox regression analysis.

The prognosis of CRT in GC patients has been reported in several previous randomized studies. The INT0116 study suggested that adjuvant CT combined with RT was effective for individuals with specific treatment modalities and disease pathological stages [12]. Findings from our study indicate that PCRT benefits patients who are age $\leq$ 65, male sex, white race, or have regional tumors. An analysis of the SEER database from 2005 to 2016 confirms earlier findings that age is a poor prognostic factor and that PCT and age greater than 60 years are related to a worse outcome. Patients who were 60 years or older had a 5-year OS that was nearly 30% lower than those who were younger. In addition, there is an ethnic disparity in PCT use and outcomes among GC patients in the United States. After controlling for patient/disease/hospital factors, race was independently associated with less PCT use [11]. There was

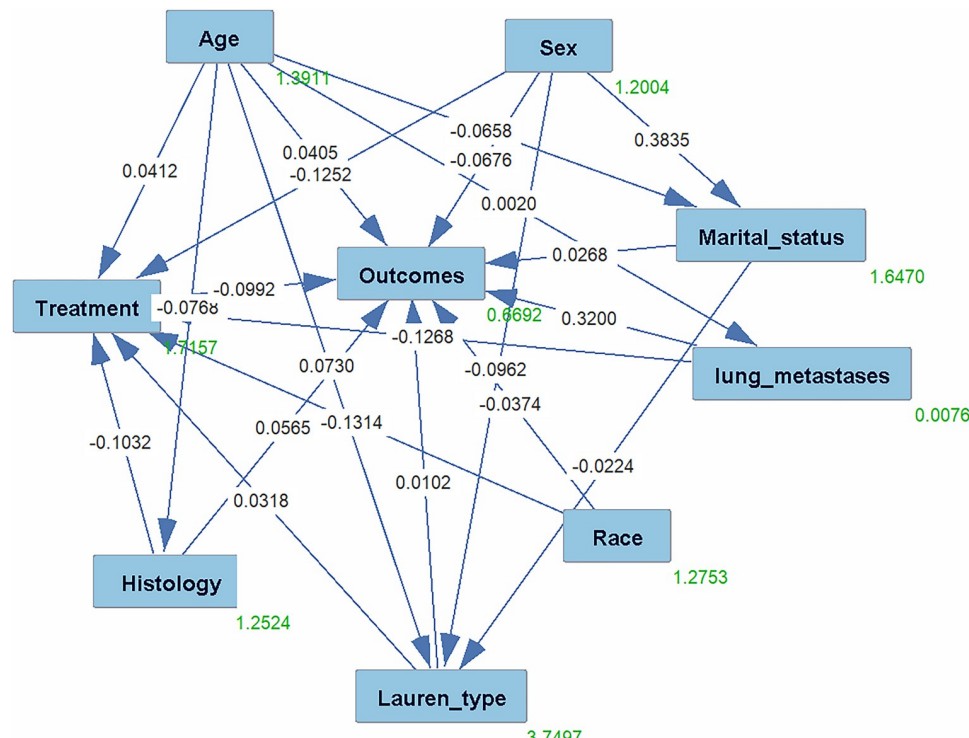

**Fig 3. Directed acyclic graphs are derived from literature and expert knowledge.** Nodes represent variables and arrows represent causal associations between variables. The Dark-colored nodes represent the factors affecting gastric cancer.

**Table 2. Advanced gastric cancer pathogenic pathway test results.**

| From | To | Vale | SE | T | P | HR |
|---|---|---|---|---|---|---|
| Race | Outcomes | -0.0374 | 0.0197 | -1.8959 | 0.058 | 1.060 |
| Sex | Outcomes | -0.0676 | 0.0314 | -2.1491 | 0.032 | 1.032 |
| Treatment | Outcomes | -0.0992 | 0.0284 | -3.4875 | <0.001 | 1.001 |
| Lung metastases | Outcomes | 0.3200 | 0.1413 | 2.2644 | 0.024 | 1.024 |
| Marital status | Outcomes | 0.0268 | 0.0116 | 2.3112 | 0.021 | 1.021 |
| Histology | Outcomes | 0.0565 | 0.0232 | 2.4322 | 0.015 | 1.015 |
| Age | Outcomes | 0.0405 | 0.0253 | 1.6004 | 0.110 | 1.116 |
| Lauren type | Outcomes | 0.0102 | 0.0167 | 0.6097 | 0.542 | 1.720 |
| Sex | Treatment | -0.1252 | 0.0287 | -4.3575 | <0.001 | 1.001 |
| Lauren type | Treatment | 0.0318 | 0.0154 | 2.0590 | 0.040 | 1.040 |
| Lung metastases | Treatment | -0.1268 | 0.1311 | -0.9672 | 0.334 | 1.396 |
| Histology | Treatment | -0.1032 | 0.0214 | -4.8220 | <0.001 | 1.001 |
| Age | Treatment | 0.0412 | 0.0235 | 1.7574 | 0.079 | 1.082 |
| Race | Treatment | -0.1314 | 0.018 | -7.3056 | <0.001 | 1.001 |

The table shows the test results of the pathogenic pathways of gastric cancer. The value represents the regression coefficient, the sign represents positive or negative correlation; SE represents standard error; T represents statistic; HR represents risk ratio.

no difference in OS between patients of different ethnicities. Another study demonstrated that GC patients' comorbidity profiles varied racially and ethnically from those of the matched cancer-free group. In particular, Whites and Blacks had higher rates of comorbid conditions than Asians or Pacific Islanders [36]. The study also discovered that anemia was the most prevalent organ, blood, and metabolic condition among non-White GC patients who were treated with CT within 6 months after diagnosis. The incidence rates were more than twice as high among Chinese, Japanese, and Filipinos as they were among Whites, Blacks, and Hispanics. The optimal therapy for locally advanced GC is perioperative multidisciplinary treatment, including PCT, RT, immunotherapy, and targeted therapy; therefore, comprehensive treatment is paramount to treatment selection [37]. In the West, PCT and PCRT are recommended for resectable patients with regional tumor invasion into the nearby tissues or middle lymph node metastasis, since their SWOG/INT-0116 trial demonstrated that the OS and recurrence-free survival (RFS) of GC patients who have had R0 resection followed by PCRT (45 Gy of RT combined with bolus fluorouracil [FU] and leucovorin) were longer than the OS and RFS of those who had surgery alone (5-year OS, 40% vs. 30%, respectively; 5-year RFS, 48% vs. 31%, respectively) [38]. These differences, such as age, race, and the presence of a regional tumor, may have an impact on the curative effect of PCRT.

The improved survival observed with PCT could be attributed to tumor regression. Multiple trials have shown that PCRT reduces the primary tumor volume and the local tumor recurrence rate and increases the rates of pathological complete response, negative margin resection, and overall survival [8, 37]. A study from South Korea showed that progression-free survival (PFS) in patients who received neoadjuvant therapy (TNT) was significantly improved when compared with that in patients who received neoadjuvant chemoradiotherapy (NCRT), and the OS tended to be longer without any side effects [39]. The Neo-PLANET study analyzed the safety and efficacy of ocrelizumab combined with CRT in the neoadjuvant treatment of locally advanced proximal gastric adenocarcinoma. The interim analysis showed a 91.7% R0 resection rate, 12 patients achieved pathological complete response (PCR, 33.3%), and the rate of major pathological response (MPR) was 41.7% [40]. Additionally, there is a potential benefit to treating micrometastatic disease and decreasing tumor cell spread following resection [8]. However, subsets of individuals might not benefit more from CRT owing to poor tolerance and limited toxicity. A study showed that up to 50% of patients who underwent surgical resection never received CRT, presumably owing to poor tolerance, whereas neoadjuvant therapy may be better tolerated and more reliably applicable [41]. In recent years, technical advances have enabled oncologists to deliver precise doses of RT and minimize exposure to critical organs, resulting in improved therapeutic effects and decreased toxicity.

For reliable causal inferences to be drawn from observational data, confounding needs to be appropriately addressed because it hides the true effect of the exposure. DAG analysis shows that age, race, and Lauren type are integral to the impact of comprehensive treatment, meaning that these factors indirectly affect the prognosis of patients with advanced-stage disease because of the direct impact on the treatment modality, so they could cause confounder bias in regression analyses. GC is considered an age-related disease, with the majority of newly diagnosed patients in the United States being over 75 years old. Elderly patients frequently have restricted inclusion in clinical trials owing to the physiological changes that occur with age, including pharmacodynamic variability, diminished organ function, and impaired functional status, which necessitate individualized treatment approaches. A study reported survival advantages in patients younger than 70 years old who received adjuvant CRT but not in those older than 70 years old [30]. Additionally, a study noted that CRT improved the outcomes of patients with the intestinal type when compared with those with the diffuse type. Although the reason for the difficulties in locoregional control of diffuse GC is yet to be discovered, the

benefit of local control may be associated with a better survival rate. These differences call for the exploration of multiple approaches in the potentially curative treatment of advanced GC. Therefore, it is possible to consider that age, race, and Lauren type may be confounding factors affecting prognosis in the advanced stage.

DAGs, a visual representation of causal assumptions, are used by researchers to better examine confounding biases related to causal questions [42, 43]. DAGs may be preferable to the conventional definition of confounding, particularly in more complicated situations, as they enable the identification of the presumptive causal mechanism and, consequently, the possibility of collider-stratification bias with certain adjustments, as well as a minimal set of factors to adjust for to remove unwanted confounding [44]. Several tutorials and reviews on the use of DAGs have been published to aid technicians and clinicians [45, 46]. Lederer et al. reviewed the general concepts, including confounding and selection bias, for researchers in pulmonary medicine [47]. By doing so, a methodical technique to deliver a summary of the context and the causal research is presented. DAGs clarify the underlying relationships and act as a visual representation of causal assumptions [48]. DAGs can thus be used to clarify confusion and suggest solutions.

Although we successfully constructed a DAG to control the bias among patients with GC after surgical resection, our study has several limitations. First, as a retrospective study, the inherent risk of selection bias is inevitable. Second, detailed treatment information is not included in the SEER database, such as the proportion of D2 lymphadenectomy, the extent of the RT field, and CT regimens. Third, although the model still performed well, there was no external validation from other larger institutions. Fourth, the accuracy of the DAGs may be compromised if a causal association between two factors is misrepresented.

## Conclusion

In conclusion, this large cohort from the SEER database revealed that PCRT has better survival benefits than PCT for GC patients with advanced-stage disease. PCRT benefits patients who are age $\leq$ 65, male, white, or have regional tumors. Hence, PCRT may be feasible for these patients. Moreover, multivariate and DAG analyses showed that sex, marital status, histologic type, lung metastases, and comprehensive treatment were associated with survival in advanced-stage patients. Furthermore, DAGs shows that age, race, and Lauren type may be confounding factors that affect prognosis in patients with advanced stage GC. DAGs, as a useful tool for contending with confounding and selection biases, are integral to the proper implementation of high-quality research.

## Supporting information

**S1 Fig. Flowchart of patient selection for this study.**
(TIF)

**S2 Fig. Feature selection using least absolute shrinkage and selection operator (LASSO) regression.** (A) Selection of tuning parameter ($\lambda$) in the LASSO regression using 10-fold cross-validation via minimum criteria. The partial likelihood binomial deviance is plotted vs log ($\lambda$). At the log ($\lambda$) of the optimal value, where features are selected, dotted vertical lines are set using the minimum criteria and the one standard error of the minimum criteria. (B) LASSO coefficient profiles for clinical features, each coefficient profile plot is produced vs log ($\lambda$) sequence. The dotted vertical line is set at the nonzero coefficients selected via 10-fold cross-validation, where sixteen nonzero coefficients are included. (C) The ROC curve is used to describe the statistical performance of the training set and test set model. (D) A nomogram for

presenting 5-year probabilities of gastric cancer patients with the advanced stage was established.
(TIF)

**S1 Table. Baseline characteristics of the radiotherapy patients between the training set and the validation set.**
(DOCX)

**S1 Data.**
(CSV)

## Acknowledgments

The authors gratefully acknowledge the efforts of the Surveillance, Epidemiology, and End Results (SEER) Program tumor registries in providing high-quality open resources for researchers. The authors would like to thank the editors and the anonymous reviewer for their valuable comments and suggestions to improve the quality of the paper.

## Author Contributions

**Data curation:** Yue Zhang.

**Funding acquisition:** HongHui Li.

**Investigation:** Juan Cao.

**Methodology:** Xiaoying Jing.

**Writing – original draft:** Cheng Zheng.

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
