## [Decision Letter · Decision Letter 0]

16 Jan 2023

PONE-D-22-32531Survival benefits of perioperative chemoradiotherapy for the advanced stage of gastric cancer based on directed acyclic graphsPLOS ONE

Dear Dr. HongHui Li,

Thank you for submitting your manuscript to PLOS ONE. After careful consideration, we feel that it has merit but does not fully meet PLOS ONE’s publication criteria as it currently stands. Therefore, we invite you to submit a revised version of the manuscript that addresses the points raised during the review process.

Your manuscript was reviewed by five reviewers. There were some suggestions and typos need further revision. Please submit your revised manuscript by Mar 02 2023 11:59PM. If you will need more time than this to complete your revisions, please reply to this message or contact the journal office at plosone@plos.org. Please include the following items when submitting your revised manuscript:A rebuttal letter that responds to each point raised by the academic editor and reviewer(s). You should upload this letter as a separate file labeled 'Response to Reviewers'.A marked-up copy of your manuscript that highlights changes made to the original version. You should upload this as a separate file labeled 'Revised Manuscript with Track Changes'.An unmarked version of your revised paper without tracked changes. You should upload this as a separate file labeled 'Manuscript'.

We look forward to receiving your revised manuscript.

Kind regards,

Wen-Wei Sung, M.D., Ph.D.

Academic Editor

PLOS ONE

Journal Requirements:

"This work was supported by Ningxia Natural Science Foundation. Item number: 2022AAC03173."

"This work was supported by Ningxia Natural Science Foundation. Item number: 2022AAC03173."

"The author reports no conflicts of interest in this work."

6. We note that you have indicated that data from this study are available upon request. PLOS only allows data to be available upon request if there are legal or ethical restrictions on sharing data publicly. For more information on unacceptable data access restrictions, please see http://journals.plos.org/plosone/s/data-availability#loc-unacceptable-data-access-restrictions. 

7. Your ethics statement should only appear in the Methods section of your manuscript. If your ethics statement is written in any section besides the Methods, please move it to the Methods section and delete it from any other section. Please ensure that your ethics statement is included in your manuscript, as the ethics statement entered into the online submission form will not be published alongside your manuscript. 

8. Please upload a new copy of Figure 2 as the detail is not clear. Please follow the link for more information: "" ext-link-type="uri" xlink:type="simple">https://blogs.plos.org/plos/2019/06/looking-good-tips-for-creating-your-plos-figures-graphics/""
"" ext-link-type="uri" xlink:type="simple">https://blogs.plos.org/plos/2019/06/looking-good-tips-for-creating-your-plos-figures-graphics/""

Reviewers' comments:

Reviewer's Responses to Questions

**Comments to the Author**

1. Is the manuscript technically sound, and do the data support the conclusions?

Reviewer #1: No

Reviewer #2: Yes

Reviewer #3: Yes

Reviewer #4: Yes

Reviewer #5: Partly

2. Has the statistical analysis been performed appropriately and rigorously? 

Reviewer #1: I Don't Know

Reviewer #2: Yes

Reviewer #3: I Don't Know

Reviewer #4: Yes

Reviewer #5: I Don't Know

3. Have the authors made all data underlying the findings in their manuscript fully available?

Reviewer #1: Yes

Reviewer #2: Yes

Reviewer #3: Yes

Reviewer #4: Yes

Reviewer #5: Yes

4. Is the manuscript presented in an intelligible fashion and written in standard English?

Reviewer #1: No

Reviewer #2: Yes

Reviewer #3: Yes

Reviewer #4: Yes

Reviewer #5: No

5. Review Comments to the Author

Reviewer #1: The authors of the manuscript "survival benefits of perioperative chemoradiotherapy for the advanced stage of gastric cancer based on directed acyclic graphs" compared perioperative chemotherapy (PCT) and perioperative chemoradiotherapy (PCRT) on outcomes in patients with locally advanced gastric cancer and investigated the usefulness of directed acyclic graphs in this setting. There is a need for thorough language editing. I feel that I did not fully understand the methods of this work.

Reviewer #2: In this manuscript, the authors reported the benefit of perioperative chemoradiotherapy for gastric cancer patients with advanced stages. They picked up the factors associated with prognosis using Cox regression and directed acyclic graphs analysis. I have only minor issues with this article.

Here are some comments from the reviewer.

1. On page 8, in line 176, are 410 patients without PCT?

2. In line 183, what CG means?

Reviewer #3: • This is a valuable work trying to enlighten the controversies in implementing perioperative chemotherapy in gastric cancer patients. The authors have used univariate, multivariate, and DAG models including various factors to be able to predict the casualty and the prognosis. I find this paper interesting but there are some points that need to be considered:

• Authors need to decide whether this paper’s objective is to approve DAG as a useful tool for comparing such outcomes or to find out the effect of PCRT. This point has to be clear in the introduction and after that, the whole manuscript storyline would change. If it is the first one, no changes are needed. But if it is the second one, please revise your introduction and conclusion.

• Introduction, line 79: I suggest starting with defining DAGs as useful tools rather than mentioning that it has been used in the current study. Also, it would be better if you could move this part completely to the methods section. Most of the audience of this paper would be clinicians that may not be interested in the statistical methodology. They might want to know the final outcome regardless of your statistical method.

• Methods, lines 140 to 152: it is very uncommon to have a review of the literature within the methods section. If the authors are seeking to come up with some rationales behind their implemented method, they should do it in the discussion. Besides, references are added in the wrong fashion. Finally, the last sentence does not indicate at all what is your solution for the abovementioned confounding factors. Please revise this part.

• Discussion: I suggest adding some justifications and assumptions based on current literature for the study’s findings. For example, why do you think male, white, and married individuals with tumor sizes of more than 5cm benefit more from PCRT?

Reviewer #4: Manuscript Type: Retrospective Study In this study, the authors explore the Pros and cons between PCT and PCRT, constructed multivariate analysis DAG model to analyze the factors affecting the prognosis of patients. As for language and narrative skills, the overall reading experience is smooth.

Reviewer #5: Dear Editor,

Thank you very much for inviting me as the reviewer of the present manuscript entitled “Survival benefits of perioperative chemoradiotherapy for the advanced stage of gastric cancer based on directed acyclic graphs”. In this study, the authors evaluated the effect of perioperative chemoradiotherapy versus chemotherapy on the survival time of patients who suffer from locally advanced gastric cancer. Moreover, using directed acrylic graphs (DAG), factors affecting survival in this group of patients have been assessed. The authors concluded that compared with perioperative chemotherapy, perioperative chemoradiotherapy was associated with longer survival among patients with locally advanced gastric cancer. Moreover, authors found that the male gender, being widowed, signet ring cell carcinoma, and lung metastases were independent factors of poor prognosis. In addition, the current study revealed that age, race, and Lauren type were among the confounding factors.

With special thanks to the authors for the valuable time and effort they put into this work, there are several major concerns regarding this manuscript. The clinical significance and causal inference method are novel, and this research can benefit PLOS ONE audience after resolving many flaws in the manuscript.

First, many major grammatical mistakes made a considerable part of the manuscript ambiguous, and the English language of the manuscript needs major revisions. Second, the use of causal inference methods, such as DAG, can face some limitations. The feature selection method will minimize the overfitting issue of DAG to some extent, but authors should avoid coming up with solid conclusions and statements. Although the authors sufficiently mentioned limitations of DAG in their “limitations”, the conclusions may mislead the audience. Third, some points are missed in the method section, and the representation of DAG (Figure 3) is not informative enough. I made several comments below which might be helpful for the authors to improve their manuscript.

According to PLOS ONE guidelines, all manuscripts should have proof that they underwent confirmation from the institutional review board. The manuscript stated “Line 101: in this study were derived from the SEER database with an openly accessible approach so the release of data did not require a local ethics approval or a statement”. I wanted to notify the Editor of this issue to act accordingly (whether this practice is acceptable or not).

1. General comments:

The article’s subject was aligned with the scope of the journal.

Ethical approval was not obtained regarding this manuscript.

1-1 Does the study presents the results of primary scientific research? Yes

1-2 Do the results reported have not been published elsewhere? No, the results of the present study are not published elsewhere.

1-3 Does the experiments, statistics, and other analyses are performed to a high technical standard and are described in sufficient detail? Partly. Since I am not an expert in casual inference analysis, I recommend inviting a statistical expert in this field.

1-4 Does the conclusions are presented in an appropriate fashion and are supported by the data? Yes.

1-5 Does the article is presented in an intelligible fashion and is written in standard English? No, many grammatical mistakes make the manuscript ambiguous for the audience. The English language of the manuscript needs major revision.

1-6 Does the research meets all applicable standards for the ethics of experimentation and research integrity? Yes.

1-7 Does the article adheres to appropriate reporting guidelines and community standards for data availability? Yes.

1-8 Manuscript and figure titles are different between the version added as metadata and the submitted manuscript. Please type the same metadata as your manuscript file.

2. In the TITLE and ABSTRACT section:

2-1 The title indicated a summary of what was done with a commonly used term, however, it needs to be more specified. Since the survival benefits of perioperative chemoradiotherapy in comparison to perioperative chemotherapy were evaluated, it is suggested to change “survival benefits of perioperative chemoradiotherapy” to “survival benefits of perioperative chemoradiotherapy versus chemotherapy”.

2-2 In the abstract, the aim is not mentioned in detail and correctly. First, please change “GC patients” to “survival of patients with locally advanced GC”. Second, it seems better to change “construct directed acyclic graphs (DAGs) to evaluate the

patient prognosis and disease risk” to “determine the factors affecting survival rate using directed acyclic graphs (DAGs)”.

2-3 Since the study just evaluated the survival rate and prognostic outcomes –not the risk of disease development-, the “disease risk of patients” located in line 32 should be removed.

2-3 In the “results” section, it is suggested to present survival rates in patients receiving PCRT or PCT, instead of “PCRT was significantly associated with the advanced GC compared to PCT for well prognosis”. Moreover, presenting a summary of baseline and clinical variables is suggested.

2-4 Since there are several grammatical mistakes, using English editing services to revise the whole manuscript to achieve standard publishing quality is strongly recommended.

3. In the INTRODUCTION section:

The background and rationale are partially disclosed. The aim of the study is mentioned.

3-1 In line 50, the authors stated that “Due to the high heterogeneity of GC in histopathology and molecular biology, the survival prognosis is dismal” which seems to be incorrect and irrelevant. Although molecular and histopathological features of tumors have a significant role in determining the patient’s survival, however, the question is, are the variation and heterogenicity of these features responsible for the dismal prognosis of gastric cancer?

3-2 In line 54, please change “advanced stage of patients” to “patients with advanced stage of GC”.

3-3 In line 67, please describe in detail the “certain benefits”. Please discuss exactly what benefits were obtained.

3-4 In line 73, please remove the word “direct” from “direct evidence”.

3-5 In line 74, the authors said that “Until now, direct evidence comparing PCT and PCRT has been limited to several small randomized trials and was mainly based on East Asian countries rather than North Americans”. Please discuss a little about the results and limitations of these previous studies.

3-6 In line 86, it seems better to change the “causal questions” to “causal relations”.

3-7 In the third paragraph of the “introduction” section, although the background, history, advantages, and importance of DAGs are fully discussed, very little is mentioned about the limitation and challenges of DAG. Please describe the issues that DAG can face (Greenland S. For and Against Methodologies: Some Perspectives on Recent Causal and Statistical Inference Debates. Eur J Epidemiol. 2017 Jan;32(1):3-20. doi: 10.1007/s10654-017-0230-6. Epub 2017 Feb 20. PMID: 28220361.).

4. In the METHODS section:

There are many ambiguities in the method section.

4-1 In the “population” section, it is recommended to describe more details about the SEER database. Moreover, it is strongly recommended to determine the type/design of the present study.

4-2 It is recommended to estimate and present the mean/median follow-up duration.

4-3 In the “Defining covariates for a directed acyclic graph” section, please describe the reasons you chose DAGs for evaluating factors that influence the GC prognosis since there are many other methods through which the effect of factors on disease prognosis and outcome can be evaluated.

4-4 In line 100, the authors said that “All the data used in this study were derived from the SEER database with an openly accessible approach, so the release of data did not require a local ethics approval or a statement.”. It is important to note that although the retrospective design of the study, de-identifying patients, and using an openly accessible database may waive the need for gathering signed informed consent from the participants, it does not relieve the need for obtaining ethical approval.

4-5 In line 104, the authors said that patients with “stage II-IV” were selected. Since there are various guidelines and criteria for staging the cancers, it is necessary to mention that in the current study/database, what criteria were used for cancer staging and what are the clinical features of patients with stage II-IV GC (regarding tumor grading, invasion, metastasis, lymph node involvement, etc.).

4-6 In line 125, please change “OS” to “overall survival (OS)”. In addition, in the whole manuscript state the abbreviations when they first appear in the text. Also prevent mentioning the full abbreviation after your first use (Referring to Line 156).

4-7 In line 141, the “The clinicopathological characteristics showed that younger patients with gastric adenocarcinoma have the more advanced and metastatic disease” seems wrong and needs correction and clarification.

5-8 In line 148, it is better to remove the “vary between races” and mention the effect of race on cancer prognosis in previous lines where demographic factors –not clinicopathological factors- are discussed –as well as the effect of the marriage status.

5-9 In line 160, what does “the most significant variables” means statistically? Please use exact statistical definitions in the “statistical analysis” section.

5-10 In line 163, please change the “Independent” to “independent”, and use correct capitalization in your whole manuscript (In line 167 “Analyses” to “analysis”, …)

5-11 In line 169, please mention the statistical significance level of the p-value.

5-12 Authors used multivariate analysis, while they did not mention how they performed such analysis and how they included variables from univariate analysis.

5. In the RESULT section:

In this section, several misreported measures need corrections.

5-1 In line 176, “A total of 1422 advanced GC patients including 289 females and 1153 males in the current study were composed of 410 patients without PCT and 1032 patients with PCRT” needs correction.

5-2 In line 179, the authors said that “In the majority of patients, 1174 patients (81.4%), the tumor was located at the antrum or pylorus”, while according to supplementary table 1, the majority of patients (n=1174) had tumors located in gastric cardiac/fundus area.

5-3 In line 181, please change the “regional” to “regional tumors”.

5-4 In line 181, please change the “GC was most frequently poor differentiation” to “The majority of gastric tumors were poorly differentiated”.

5-5 In line 204, what does “visible heterogenicity” mean?

5-6 In line 215, “Notably, PCRT was still significantly associated with the advanced GC for well prognosis” needs correction. It is suggested to use this: “PCRT was still significantly associated with a better survival in patients with advanced GC”.

5-7 many parts are ambiguous and meaningless:

In line 219, the “show the possible relationship with advanced GC patients” is wrong and meaningless and needs correction.

In line 221, “cause the path of each variable in the advanced GC, it does not mirror which pathway is significant” is wrong and meaningless and needs correction.

In line 222, “The prognosis of sex, marital status, histology type, lung metastases, and treatment were significantly associated with advanced GC in both cox regression models and DAG tests” is wrong and meaningless and needs correction.

In line 225, the “Additionally, DAGs results showed that the factors of sex, race, histology type and Lauren type impact the effect to administer treatment” is wrong and meaningless and needs correction.

6. In the DISCUSSION and CONCLUSION section:

The findings are partially discussed and explained. However, the potential reasons accounting for the similarities and differences between the results of the current study and previously published studies are not well explained. The limitations are partially disclosed. The conclusion is aligned with the findings and aim of the study.

6-1 The discussions should be rewritten since it is monotonous and not informative. There are many mistakes, including:

In line 239, “However, whether PCRT is more survival benefits than PCT in resectable GC patients remains unclear” is wrong and needs correction.

In line 242, it is recommended to change “the specific clinical characteristic” to “demographic and clinical characteristics”.

In line 247, please exactly describe the “treatment modalities” which have been evaluated in the mentioned study.

In line 252, please exactly describe the “specific treatment modalities and disease pathological stages” which have been evaluated in the mentioned study.

In line 23, please change the “deficiencies” to “limitations”.

In line 266, please change the “reduction in” to “reduces”.

In line 267, the authors said that PCRT was associated with “local tumor recurrence” in previous studies. Since local tumor recurrence is considered an adverse outcome, it seems that this sentence is wrong and needs correction. Please again check the findings of the mentioned studies and revise this sentence.

In line 305, please change the “bise” to “bias”.

5-2 Please describe the potential reasons which may be responsible for reaching a better survival following the implementation of PCRT than PCT.

5-3 In line 320, please mention the exact association between the mentioned variables and survival.

7. Tables and Figures:

The tables do not have appropriate titles.

7-1 In Table 1, there are many mistakes in presenting the numbers/sizes of each group/variable. Are the numbers in the parenthesis showing the percentage?

7-2 In Table 1, the sum of participants in subgroups does not match the total participant in many cases. Please add a row for missing values, if it is the case.

7-3 In Figure 3, the structure of the variables is ambiguous. Please re-draw your figure with a more informative structure. You can find the structure of the DAG in the following paper useful (https://doi.org/10.1002/cncr.31155).

8. References:

8-1 A majority of references are old (do not belong to the recent 10 years). It is strongly suggested to use references that are published in recent 10 years if possible. Some suggestions for enriching your mansucrit:

Nishikawa G, Banik P, Thawani R, Kardosh A, Wood SG, Nabavizadeh N, Chen EY. Comparison of neoadjuvant regimens for resectable gastroesophageal junction cancer: a systematic review of randomized clinical trials across three decades. J Gastrointest Oncol. 2022 Jun;13(3):1454-1466. doi: 10.21037/jgo-22-29. PMID: 35837173; PMCID: PMC9274047.

Grassadonia A, De Luca A, Carletti E, Vici P, Di Lisa FS, Filomeno L, Cicero G, De Lellis L, Veschi S, Florio R, Brocco D, Alberti S, Cama A, Tinari N. Optimizing the Choice for Adjuvant Chemotherapy in Gastric Cancer. Cancers (Basel). 2022 Sep 25;14(19):4670. doi: 10.3390/cancers14194670. PMID: 36230592; PMCID: PMC9563297.

9- Funding and Conflict of Interest

Funding and conflict of interest are well declared.

9-1 Please add the initials of the funding receipting in your Funding statement.

10- Reviewer Conflict of Interest (Only for Editor)

I declare to have no conflict of interest with the manuscript or the authors and had not been working with any of the authors named in this manuscript for at least during past 36 months. I declare to preserve the confidentiality and rights of the presenting manuscript. All suggested articles are relevant to the topic, and I have no conflict of interest with the recommended articles. My mentee at my research institute (Zohreh Tajabadi) helped me for evaluation of this manuscript and provided suggestions to improve English language of this work.

Best Regards,

Reviewer

6. PLOS authors have the option to publish the peer review history of their article (what does this mean?). If published, this will include your full peer review and any attached files.

Reviewer #1: No

Reviewer #2: No

Reviewer #3: **Yes: **Mohammadreza Azangou-Khyavy

Reviewer #4: No

Reviewer #5: No

---

## [Author Response · Author response to Decision Letter 0]

10 Feb 2023

Dear reviewers

We thank the reviewers for the time and effort that they have put into reviewing the previous version of the manuscript. Their suggestions have enabled us to improve our work. Appended to this letter is our point-by-point response to the comments raised by the reviewers. We have now worked on both language and readability and have sought help from AJE's English editing services to revise the whole manuscript to achieve standard publishing quality. All changes made to the manuscript in response to the reviewer’s comments are highlighted in red. The page and line numbers mentioned in this response refer to those numbers in the revised manuscript. 

We hope that the revised manuscript is accepted for publication in the Journal of PLOS ONE

Sincerely,

Cheng Zheng

---

## [Decision Letter · Decision Letter 1]

20 Mar 2023

Survival benefits of perioperative chemoradiotherapy versus chemotherapy for advanced stage gastric cancer based on directed acyclic graphs

PONE-D-22-32531R1

Dear Dr. HongHui Li,

We’re pleased to inform you that your manuscript has been judged scientifically suitable for publication and will be formally accepted for publication once it meets all outstanding technical requirements.

Kind regards,

Wen-Wei Sung, M.D., Ph.D.

Academic Editor

PLOS ONE

Reviewers' comments:

Reviewer's Responses to Questions

**Comments to the Author**

1. If the authors have adequately addressed your comments raised in a previous round of review and you feel that this manuscript is now acceptable for publication, you may indicate that here to bypass the “Comments to the Author” section, enter your conflict of interest statement in the “Confidential to Editor” section, and submit your "Accept" recommendation.

Reviewer #3: (No Response)

2. Is the manuscript technically sound, and do the data support the conclusions?

Reviewer #3: (No Response)

3. Has the statistical analysis been performed appropriately and rigorously? 

Reviewer #3: (No Response)

4. Have the authors made all data underlying the findings in their manuscript fully available?

Reviewer #3: (No Response)

5. Is the manuscript presented in an intelligible fashion and written in standard English?

Reviewer #3: (No Response)

6. Review Comments to the Author

Reviewer #3: (No Response)

7. PLOS authors have the option to publish the peer review history of their article (what does this mean?). If published, this will include your full peer review and any attached files.

Reviewer #3: **Yes: **Mohammadreza Azangou-Khyavy

---

## [Editor Report · Acceptance letter]

6 Apr 2023

PONE-D-22-32531R1 

Survival benefits of perioperative chemoradiotherapy versus chemotherapy for advanced stage gastric cancer based on directed acyclic graphs 

Dear Dr. Li:

I'm pleased to inform you that your manuscript has been deemed suitable for publication in PLOS ONE. Congratulations! Your manuscript is now with our production department. 

Kind regards, 

on behalf of

Dr. Wen-Wei Sung 

Academic Editor

PLOS ONE